# Go-Go Music and Racial Justice in Washington, DC

Collin Michael Sibley

Department of Religious Studies, University of California, Santa Barbara, CA 93106, USA; cmsibley@ucsb.edu

**Abstract:** In 2019, a noise complaint from a new, white resident of Shaw, a historically Black neighborhood of Washington, DC, led a local MetroPCS store to mute the go-go music that the storefront had played on its outdoor speakers for decades. The cultural and social implications of muting go-go music, a DC-originated genre of music that has played a central role in DC Black culture, inspired a viral hashtag, #dontmutedc, on social media, as well as a series of high-profile public protests against the muting. The #dontmutedc protests highlighted the increasing impact of gentrification on DC's Black communities, and connected gentrification to several other important social issues affecting Black DC residents. In the wake of the #dontmutedc incident, several DC-area activist organizations have integrated go-go music into major, public-facing racial justice projects. The first part of this article presents a brief history of go-go music and race in DC community life, mainstream media, and law enforcement in order to contextualize the work of go-go-centered activist work in the aftermath of the #dontmutedc protests. The second part of this article highlights the go-go-centered activist work of three organizations: the Don't Mute DC movement, Long Live Go-Go, and the Go-Go Museum and Café. These movements' projects will be used to categorize three distinct approaches to go-go-centered racial justice activism in the Washington, DC, area.

**Keywords:** go-go music; racial justice; Don't Mute DC; Long Live Go-Go; Moechella; Black studies; Washington DC; DC statehood; public protest

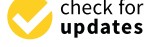



## 1. Introduction

In 2019, a new, white resident of Shaw, a historically Black neighborhood of Washington, DC, filed a noise complaint against a local MetroPCS store for playing music on its outdoor speakers. This storefront on the corner of 7th St. and Georgia Ave., NW, had been playing go-go music, a percussion-heavy, DC-originated musical style central to DC Black culture, on its outdoor speakers for years prior to this new resident's noise complaint. In response to the complaint, the MetroPCS store turned off the music on its outdoor speakers. Frustrated with this situation, which seemed deeply symbolic of the cultural and social changes caused by the rapid 21st-century gentrification of the District, Howard University student Julien Broomfield coined the hashtag #dontmutedc on Twitter to protest the incident. The hashtag and the events that inspired it quickly gained regional attention, inspiring a firestorm of social media activity and a large in-person public demonstration at the corner of 7th St. and Georgia Ave. NW. The scholar Natalie Hopkinson and local community activist Ronald Moten authored an online petition to turn the go-go music at the Metro PCS storefront back on, a petition that gained over 80,000 signatures by the end of the day. The next evening, Moten and Justin "Yaddiya" Johnson, a DC-based activist, rapper, and musical promoter, organized a protest against the ban at the corner of 14th and U St. NW, the same intersection where major riots erupted in 1968 to protest the assassination of Martin Luther King, Jr. The next day, the CEO of T-Mobile, MetroPCS's parent company, personally guaranteed that the go-go music at the storefront would be turned back on, and the music was indeed turned on following a press conference hours later.

The flurry of public outrage and activist action inspired by the Don't Mute DC incident demonstrates several important truths—features of the connection between go-go and racial activism in 21st-century Washington, DC. First, for many Black DC-area natives, go-go is

more than just a style of music: it is also a repository of DC-area Black cultural identity and history. Go-go bands, musicians, venues, performances, and recordings ("PA tapes") have served as cultural touchstones in the DC area for decades. The muting of the go-go music that had played at the same Shaw storefront for decades seemed to Don't Mute DC protestors to be more than a simple dispute over noise: rather, they saw the incident as a continuation of the racialized social and cultural marginalization of go-go in its own home city. Racially charged scrutiny from local media and law enforcement and the socioeconomic pressures of gentrification on DC's Black communities have dogged the go-go scene for decades. The increasingly rapid pace of DC's gentrification in the 21st century has caused rapid increases in rent and cost of living, leading many of DC's Black residents into deep economic insecurity. High rent costs and corporate land acquisition across the District have also radically changed the racial and socioeconomic demographics of formerly Black-majority neighborhoods and forced many Black longtime residents to move to the suburbs or out of the area entirely. DC's official status as a federal territory of the United States rather than a U.S. state severely limits DC residents' voice in major features of DC public policy, curtailing the legislative channels by which DC residents can curb the effects of gentrification on DC neighborhoods.

Historically, go-go music has prioritized energy, danceability, and audience participation over sustained political or social commentary. In the wake of the #dontmutedc protests, however, go-go music has emerged as a powerful symbol for public activism in 21st-century Washington, DC (Wartofsky 2020). Multiple regional activist movements have found ways to connect go-go music to political, social, and cultural causes in the DC area. The work of these organizations has collectively boosted the public presence of go-go in the DC area and drawn attention to political and social issues such as DC statehood and police violence. This article will highlight three recent activist mobilizations of go-go music—Don't Mute DC, Long Live Go-Go, and the Go-Go Museum—and compare their approaches in integrating go-go into public-facing activist projects in the DC area.

## 2. Go-Go and "Chocolate City"

Washington, DC, has a long history as a major Black population center and Black cultural hub in the Mid-Atlantic region of the East Coast. Following the Emancipation Proclamation, DC received a large influx of newly freed Black folks from Southern States, adding to its already substantial Black population. Federal government positions offered excellent job stability and potential for professional advancement for Black workers compared to other industries, allowing for the creation of a sizable Black middle class in the DC area. Residential segregation also created a number of Black-majority neighborhoods in the District. By the 1920s, one of these neighborhoods, the U Street corridor in Northwest DC, had become known as the "Harlem of the South" due to the vitality of its Black cultural production. Internationally famous artists such as Duke Ellington began their careers in the U Street scene before moving to New York's Harlem. By the 1970s, the District of Columbia had become a majority-Black city, famously inspiring the funk group Parliament's 1975 song, "Chocolate City".

The guitarist and singer Chuck Brown is universally recognized as the founder (popularly, the Godfather) of go-go. A child of a poor single mother in eastern Maryland, Brown cut his teeth as a performer while incarcerated as a young adult at Lorton Correctional Facility in Northern Virginia. Following his release from prison, Brown joined Los Latinos, a Salvadoran band in the DC area. Brown credited his time playing with Los Latinos in the mid-60s with introducing him to the conga drum, which went on to become a core instrument in go-go bands and one of the most recognizable features of go-go's sound.[1] When Brown split off in 1966 to form his own band, Chuck Brown and the Soul Searchers, Brown used conga drums to anchor the percussion grooves that he used to fill the space between songs. This format became characteristic of Chuck Brown's shows and became the standard format of live go-go performances. Chuck Brown's magnetism as a performer

and relatability to a wide range of DC audiences helped him popularize go-go as a genre distinct from the soul and funk traditions in which Brown had trained.

By the 1980s, go-go music had come to sustain an entire regional industry of Black musicians, promoters, venue owners, and music sellers. Go-go bands like Chuck Brown and The Soul Searchers, Rare Essence, Trouble Funk, and Experience Unlimited (E.U.) each garnered large loyal followings that regularly showed up for shows, and a number of other bands carved out successful niches in the scene. In the 1980s, go-go shows ("go-gos) became runways for DC's burgeoning streetwear culture, which earned fame on the East Coast for innovative applications of mainstream fashion brands like New Balance and Gucci. Go-go patrons commonly wore DC-area streetwear brands like Madness ALLDAZ and HOBO to represent the specific region of the District or adjacent Prince George's County they had come from.[2] In addition to official band merchandise, audio cassettes ("PA tapes") of go-go shows became popular local commodities sold by a local industry of third-party tape (and later, CD) sellers. In the 1990s, a new, more percussive sound of go-go, associated with groups such as Junkyard Band, Northeast Groovers, and Backyard Band, became the voice of the next generation of go-go fans. The next major stylistic wave came with the energetic, rototom-dominated bounce beat sound of the 2010s, pioneered by groups like TCB, XIB, TOB, and others (Richards 2012). Although far fewer go-gos are held today than in their heyday in the 1980s and 90s, go-gos remain popular with both older and younger generations of Black DC-area residents.

In recent decades, go-go has seen a resurgence in its public profile in the DC area. In addition to go-go's enshrinement as the official music of Washington, DC, large-scale public performances of go-go have been on the rise. Over the last several years, the NFL's Washington Commanders, NBA's Washington Wizards, and NBA G-League's Capital City Go-Go—named for the musical genre—have all featured live go-go performances as part of game-day programming. Many of these performances have also featured Beat Ya Feet dancing, a style of dancing invented in the go-go scene. The District has also funded several public performances by notable groups, including a 2021 Rare Essence concert on the rooftop of the MLK Library in downtown DC.

Compared to other contemporary genres of American music, go-go has been especially rooted in live performance. In contrast to popular genres like hip-hop, rock, and R&B, recordings of go-go music are principally taken from live shows, rather than dedicated studio sessions. Live performances remain the primary medium by which go-go is consumed in the DC area, and several important features of the genre itself rely on this live performance medium. Call-outs—on-mic recognitions of people in attendance—are nearly ubiquitous even in recorded go-go music. Similarly, vocals from the audience, whether in a call-and-response format or singing along with the song, are almost universal even in the recorded formats (Lornell and Stephenson 2009).

Natalie Hopkinson (Hopkinson 2012) has argued that go-go's roots in live performance have allowed the genre to stay somewhat counter-cultural, relative to mainstream American culture, while parallel genres of Black music like hip-hop have become fully integrated into the commercial infrastructure of American popular music.[3] Hopkinson has described the go-go as a Habermasian public, where attendees exchange news and perspectives on community events.[4] The improvisational role of the talker, a central member of the band who directs the ensemble and provides ad-lib commentary and vocal shout-outs over the band's percussion pockets, allows the bands themselves to direct or participate in these exchanges during the shows themselves. Much of what a talker says during a go-go depends on the specific crowd in attendance: a talker's shoutouts recognize both individual fans and street crews,[5] and sometimes engage in short vocal exchanges with them. PA tapes of shows document these exchanges and circulate them to an even broader audience of fans and consumers.

### 3. Go-Go and Race in the Public Eye

As a genre most popular in lower-income Black communities, go-go has been persistently racialized on both the local and national levels. This racialization has taken various, complementary forms in both mass media and DC public administration.

The Hollywood film *Good to Go*, produced by the Jamaican director Chris Blackwell, offers a prototypical example of mainstream American racial attitudes towards the go-go scene.[6] *Good to Go*, originally billed as a high-budget, full-length filmic introduction to DC's thriving go-go scene, was a major critical and commercial flop both regionally and nationally. Rather than painting a realistic picture of the 1980s go-go scene, *Good to Go* features several well-worn racial tropes of both go-go music in local media coverage and Black music in contemporary American cinema. The main character of the film, played by the folk singer Art Garfunkel, is an intrepid white reporter for a major DC newspaper who begins covering the go-go scene after hearing about the go-go related rape and murder of a DC-area woman. Garfunkel's character discovers that this report had been fabricated to frame a conga player on the go-go scene. After a plot line conspicuously featuring brutal violence, hard drug use, and robbery sprees, Garfunkel's character collaborates with members of the go-go scene to prove the conga player's innocence and indict the crooked detective who framed him.

Even though *Good to Go* was presented by writers as a critique of biased media coverage of go-go, local audiences overwhelmingly received the film as a rehashing of these stereotypes, rather than a thoughtful reversal of them. *Good to Go*'s portrayal of the go-go scene, although caricatured, maps onto the ways that news media, city officials, and law enforcement have approached the real-life go-go scene. In the 1980s and 1990s, local news media sensationalized violent incidents at or around go-gos as "go-go killings". Many of these "go-go killings" were not meaningfully connected to go-gos themselves; they sometimes occurred multiple blocks away from performance venues or hours before or after the performances themselves. For instance, a *Washington Post* article on a 1987 mass shooting following a Rare Essence concert spends ample time describing community concerns about go-go music's association with violence and alcohol-fueled disorder. The shooting itself, however, occurred outside of the venue while the band was packing up, and had no discernible connection to the band or performance itself (Stevenson and Fisher 1987).[7] Media coverage sensationalizing the violence and disorder associated with the go-go scene both amplified popular associations between go-go and criminal disorder and deflected attention from the structural problems contributing to violence in the first place.

The association of go-go with criminal disorder in popular media continues to affect the District of Columbia's administrative relationship with go-go bands, venues, and patrons. Similar to the way in which Cabaret Laws in New York City impinged on Black jazz clubs in the 1940s and 1950s, go-go venues consistently face difficulties securing permits from DC government due to go-go's stereotypical association with disorder, drug use, late-night noise, and violence. Go-go venues have a particular challenge obtaining and retaining liquor licenses because of city officials or local business owners' protests that liquor licenses will contribute to rowdy behavior.[8] Given go-go's prototypical association with nightlife, losing access to liquor licenses dramatically affects the long-term survival prospects of go-go venues. City officials have also cited noise complaints as a reason to restrict the neighborhoods where go-go venues may be issued permits. DC police have even boasted about maintaining a "Go-Go Report" with a nightly list of local go-gos in the region that allows them to anticipate sites of disorder or criminal activity in advance, easing police deployment.[9] This report is a remarkably literal expression of an attitude that has pervaded multiple levels of local government and law enforcement. Similar to media coverage highlighting "go-go killings", police action targeting go-go bands and venues places some degree of criminal responsibility on the go-go scene itself, and accordingly discounts other contributing factors.

## 4. Three Pathways to Go-Go Activism

The #dontmutedc protests, as well as the nationwide activist trends associated with the Black Lives Matter movement, have amplified the potential connections between go-go and contemporary Black cultural and political issues. In the years following the #dontmutedc protests, several movements and organizations have solidified these connections.

### 4.1. Don't Mute DC

Following the #dontmutedc protests, the authors of the Don't Mute DC petition, Natalie Hopkinson and Ronald Moten, founded an organization, Don't Mute DC, aimed at furthering legislative policy advocacy on causes related to go-go and DC's Black communities. In 2020, Don't Mute DC issued a report informed by recommendations collected from community members at a 2019 "Don't Mute DC Call to Action Conference", a town hall, an online survey, and organizations such as Howard University, the Smithsonian Center for Folklife and Culture, and the National Endowment for the Arts.[10] Don't Mute DC's 2020 report outlines a number of policy recommendations related to go-go, including the establishment of a centralized archive of go-go recordings and the introduction of go-go-centered education into DC curricula.

The Don't Mute DC movement's 2020 report (Stephenson and Lea 2020) and other activist activities exemplify a legislative approach to integrating go-go into public activism. This approach has the advantage of being able to advocate directly for long-term policy focuses addressing structural issues affecting the go-go scene and Black DC more broadly. While the District Government has occasionally shown an interest in go-go—most notably, through Mayor Marion Barry's Showmobile program—Mayor Bowser's declaration of go-go as DC's official music suggests the possibility of a more sustained legislative relationship between go-go and DC public policy. Broadly speaking, Don't Mute DC's action report envisions a reciprocal relationship between investments in the go-go scene and policies in other dimensions of DC cultural, social, and political life. The report describes go-go music as a "vital, irreplaceable natural resource" that is not only a key feature of DC's cultural history, but also is a force that inspires community action.

In late 2023, Ronald Moten announced that Don't Mute DC will stage a series of public protests against Washington Wizards and Washington Capitals owner Ted Leonsis's proposal to move the teams' arena out of the District into a new stadium complex in Northern Virginia. The Wizards and Capitals' current arena, Capital One Arena, has been credited with the economic revitalization of the Chinatown neighborhood of DC, and some city officials and community figures have expressed fears that moving the teams out of the neighborhood will be a massive hit to local businesses. Additionally, the symbolism of moving Washington DC sports teams from a historically Black-majority city to the white-majority suburbs of Northern Virginia has, like the muting of go-go in Shaw, read to many DC residents as an implicit cultural statement. Like other Don't Mute DC actions, Don't Mute DC's protests are centered on a specific policy issue and have a set of concrete demands.

### 4.2. Moechella and Long Live Go-Go

In 2019, immediately following the #dontmutedc incident, Justin "Yaddiya" Johnson—a DC-based rapper, activist, and musical promoter—began organizing public protest events centered on live performances of go-go music. Community members dubbed these events "Moechellas", a playful portmanteau of the regional DC Black slang term "moe" ("dude", "bro") and Coachella, the famous corporate music festival held annually in Indio, California. Between 2019 and 2022, Yaddiya and his Long Live Go-Go organization hosted many Moechella events, each centered on a social, political, or cultural topic, such as rising DC housing costs, racially biased police violence, and the fight for DC statehood. Moechellas' large turnouts led to wide coverage on local news networks, further amplifying the reach of the movement's protest actions and activist platform.

In 2023, Coachella sued Yaddiya and Kelsye Adams, Long Live Go-Go Art Director and DC voting rights advocate, for infringing on Coachella's copyright of the Coachella name and brandmark, although Yaddiya had already agreed to a previous request from Coachella to change the Moechella name and Moechella branding on his merchandise (Weiner 2023). In addition to arguing that Long Live Go-Go had profited off the sale of merchandise featuring the word "Moechella" in Coachella-style font, Coachella's lawsuit also mentioned a shooting "at" a Moechella event in which a fifteen-year-old boy was killed and three other people were wounded. Coachella's lawyers argued that Moechella's name implied an association with Coachella that harmed Coachella's brand in light of this incident, and that Moechella had "intentionally traded" on Coachella's "goodwill" (Weiner 2023).

Coachella's lawsuit against Moechella misrepresented important details about the shootings and the broader Moechella movement. Although Coachella's legal team associated the shootings with Moechella, the shootings did not occur at the performance itself, but rather nearby and after the day's musical programming had already ended.[11] Moreover, none of the people involved had any connection to the Moechella event, its organizers, or the musical performers featured. This incident came after four years of fully peaceful Moechella demonstrations, further calling into question whether the shootings could be attributed to the impact of the movement itself (Jordan 2022). In the same way that local media have historically sensationalized violence at or near go-go venues as "go-go killings", Coachella's legal team presented the event near 14th and U as an event for which Moechella is legally responsible, at least to the extent that it damaged Coachella's brand. This move adheres to the same racialized logic that has long connected go-go music with criminality and public disorder (Wartofsky 2022).

Coachella's lawsuit had a devastating effect on Long Live Go-Go's online presence. Because of Coachella's copyright infringement claims, Long Live Go-Go's official Instagram and Twitter pages were taken down, stripping the movement of two of its primary outlets of organizational communication. In reaction to the lawsuit and its impacts on the organization, Long Live Go-Go shifted its attention to multimedia and other culturally focused projects, documenting both Moechella itself and go-go's cultural impact in DC more broadly. In addition to an already-published photo essay compilation, Long Live Go-Go is producing an official video documentary on Moechella, "Save the Movement", that will highlight both the protests themselves and the social and cultural issues that motivated their attendees to participate.[12] In late 2023, Long Live Go-Go launched a "Go-Go Alumni" media campaign highlighting important figures in DC cultural life (Beaujon 2023). The campaign features photo shoots of these figures, including Ben's Chili Bowl co-founder Virginia Ali and director and New Balance designer June Sanders, dressed in a signature, college-style "Go-Go Alumni" hoodie designed by Kelsye Adams. Long Live Go-Go has also begun operating a pop-up bar, The Alumni Spot, featuring go-go and DC-themed cocktails and food, go-go and other musical performances, and framed photos of "Go-Go Alumni" honorees. The Go-Go Alumni campaign and its expansion into a pop-up venue both reflect Long Live Go-Go's focus on documenting DC cultural history and featuring it in public facing culture production. Beyond these projects, Long Live Go-Go continues to organize talks and other events, and also has offered musical support for relevant protest events co-hosted by other organizations (such as DC Vote's 2023 Rally for DC Statehood). The organization has also provided support for District-based cultural events, such as September 2023's Art All Night.

The Moechella and Long Live Go-Go model of go-go activism combines political activism with live go-go performances. While Don't Mute DC has pursued the reform of these issues through official legislative channels, Long Live Go-Go has prioritized mass protest events and cultural programming directly powered by community action. Moechellas' large-scale public go-go performances on DC city streets serve as powerful symbolic and de facto assertions of go-go's right to space in the District's public soundscape. Moechellas also use the continuing regional popularity of go-go music to draw large,

energetic crowds to activist demonstrations and events. Moechellas have featured popular local bands and musicians, whose fame further raises the profile of Moechellas and draws fans out to demonstrations. As an energetic, highly danceable genre of music, go-go music is a natural fit for public protest. Go-go bands have ample experience winding up crowds, and talkers use on-mic commentary both to direct protesters in real time and to deliver activist messages to the assembled crowds. Go-go's signature percussion instruments are also well-suited to the acoustics of outdoor protests, since their sound carries far even without extensive amplification.

## 5. The Go-Go Museum and Café

The Go-Go Museum and Café, set to open in 19 February 2024, is a project overseen by Ronald Moten, co-founder of the Don't Mute DC organization and a longtime DC-area activist. The Go-Go Museum and Café is located in the Anacostia neighborhood of southeast DC, a major historical Black population center in the District. The project has been funded by a combination of private donations and a USD 50,000 grant from the DC government (Williams 2023). The nearly 3000-square-foot site will feature digital and physical museum exhibits on go-go history, a stage for live go-go performances, and a café with an eclectic menu combining African, Caribbean, Latin, and local DC culinary influences to symbolize the various regional musical influences reflected in go-go music (Thomas 2023). Although the physical Go-Go Museum and Café has not yet opened, the Go-Go Museum has sponsored several educational lectures and community events focused on go-go history and community dialogue. Moten has also procured a "mobile go-go museum", a 29-foot-tall bus featuring a small digital exhibit space and a rooftop performance stage (Thomas 2023).

The Go-Go Museum and Café project exemplifies a venue-based approach to go-go activism. Creating a permanent museum space dedicated to go-go, a historically Black and sociopolitically marginalized musical genre, is impactful given DC's rapid gentrification in the 21st century, which has displaced longstanding Black communities in the district and severely increased socioeconomic pressures on the District's low-income Black residents. Go-go performance venues have historically had trouble opening and staying open, in large part because of discriminatory DC legislative and policing policies recapitulating the racialized association between go-go and criminality or disorder. Billed as a museum space and community center, the Go-Go Museum avoids some of the public relations and image issues that have affected the survival of many other physical sites associated with go-go music. Both the programming and architecture of the museum discourage the formation of large, unruly crowds, deflecting some of the crime-related concerns that both DC Police and present or future community businesses or residents may levy about a permanent go-go venue. Even though the stereotypical association of go-go with crime and disorder is itself deeply and directly informed by racialized mainstream American attitudes towards urban Black culture, sidestepping these stereotypes has an undeniable pragmatic advantage, particularly in the light of Republican Congresspeople's recent politically motivated interventions in local DC policy. As a permanently-open museum space, the Go-Go Museum is also well-suited to serve as an educational resource on go-go geared towards both children and adults.

## 6. Conclusions

The challenges facing both go-go music and the Black communities of Washington, DC, are too great for a single organization to undertake on its own. Each of the models of go-go activism exemplified by the organizations described above can make a particular set of contributions to local policy and discourse. In order to harness go-go's full potential as a tool of racial justice activism, multiple activist pathways must be able to coexist. At the same time, go-go-centered activism must be careful to avoid symbolic co-option (and subsequent derailment) by elements of the political establishment that are not necessarily committed to uplifting DC's Black communities. This is especially true of activist work that

relies on formal political or legislative action, given that DC residents' political agency is severely limited by DC's legislative status as a federal U.S. territory.

Go-go-centered activism is an approach to racial justice work that builds on regionally important features of DC's cultural and racial history. In this sense, go-go-centered activism relates to other race-centered social justice movements in other cultural contexts across the world. Go-go music provides a cultural grounding for go-go activism and charts pathways to public action. The regional specificity of many features of go-go activism also offers an important reminder of the pivotal role that local history and culture can play in responses to broader structures of racial and cultural oppression.

**Funding:** This research received no external funding.

**Institutional Review Board Statement:** Not applicable.

**Informed Consent Statement:** Not applicable.

**Data Availability Statement:** Data are contained within the article.

**Conflicts of Interest:** The author declares no conflict of interest.

## Notes

[1]   Hopkinson, Go-Go Live, p. 39.
[2]   Hopkinson, Go-Go Live, p. 42.
[3]   Hopkinson, Go-Go Live, pp. 153–55.
[4]   Hopkinson, Go-Go Live, pp. 8–9.
[5]   In DC culture, a crew is a group of friends and family associated with a particular street corner or housing development.
[6]   For a fuller treatment of the production, reception, and cultural politics of Good to Go, cf. Lornell and Stephenson, The Beat, pp. 209–18.
[7]   For another pertinent example of Washington Post coverage of go-go killings, see (Sanchez 1987).
[8]   Cf. Chapter 2, "Club U" in Natalie Hopkinson's Go-Go Live for a case study highlighting these issues.
[9]   Hopkinson, Go-Go Live, pp. 3–4.
[10]  "Don't Mute Go-Go DC Plan", p. 5.
[11]  (Jordan 2022). The Moechella in question was a 2022 day-long Juneteenth celebration along U St. NW, a historic center of Black culture and nightlife in the District. The shootings occurred near the intersection of 14th and U NW while musicians were packing up equipment and breaking down sets.
[12]  Organizational communication, Mar. 2023.

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
