# Peer review of "Go-Go Music and Racial Justice in Washington, DC"

_genealogy, doi:10.3390/genealogy8010009_

Round 1
Reviewer 1 Report
Comments and Suggestions for Authors
Original paper and important subject underrepresented in the literature. Very well-written and organized--allowed for a clear grasp of the concepts and central argument. Nice flow to the paper and solid contextual grounding. One of the more informative and well-executed pieces I've read recently.
Author Response
Thank you so much for your kind words and for taking the time to read my manuscript!
Reviewer 2 Report
Comments and Suggestions for Authors
There is lots to love about this essay, which shows how the long history of DC's Go-Go music scene has flourished over the years despite a complex relationship to the community in terms of race, (Black) subcultural music forms, and the increasing gentrification of DC over the last 20 years. [Note: I attended graduate school in DC in the early 2000s and live close enough to DC presently to visit a few times a year. The changes to the city during this time have been overwhelming.] The writer(s) use the #DontMuteDC movement from 2019 as a entry way into a discussion about how Go-Go music is being weaved into the DC's community and how it is being used increasingly to mobilize political action. Drawing off the work of Hopkinson and others, the essay does good work laying out a basic history of the music, its culture, and how it has collided with DC's varied "interests" (racial, class, political, and otherwise) over the years.
A few things would make this piece read a bit better. Right now, the thesis doesn't appear until half-way through the paper. Consider moving it to the end of the first section (above Go-Go and "Chocolate City"). The last two paragraphs of the Good to Go section could be strengthened with some light primary source research (which could be included as footnotes) presenting examples of when the DC local press considered things "Go-go killings" when they actually took place somewhere far from actual Go-Go shows. Connecting the liquor license issue with predominantly Black jazz clubs in New York City in the 40s and 50s ("Cabaret Laws") might be good way of illustrating this sustained effort by the police and licensing boards to criminalize Black music cultures. Lastly, it should be noted that Parliament's song "Chocolate City" was released in 1975, not in the 1960s.
The last sections end the essay strongly. The case of Moechella is a fascinating one, one showing how even big business is using its power to stop Black cultural expression. Ultimately, I really like this piece, but consider some re-organizing to make it flow better and some added detail to connect the history of Go-Go's marginalization within its own city will give it a bit more oomph.
Author Response
Thank you for your thoughtful feedback on my paper. I have tried to act on all of your suggestions here. Per your recommendation, I have bolstered the thesis statement in the first section of the paper. I have also removed the paragraph on DC statehood to improve the flow of the introductory paragraphs and help the thesis itself stand out. I have added citations of two Washington Post articles describing "go-go killings", and I have unpacked one of these articles in the main text of my piece. I have included a brief reference to the NYC Cabaret Laws to help interested readers see the continuity with other ways city governments have regulated Black performance venues. Finally, I have corrected the release date of "Chocolate City". A personal thanks for catching that error of mine- I never would have forgiven myself if that had made it to print!
Reviewer 3 Report
Comments and Suggestions for Authors
Please see attached file.

Author Response
Thank you for your thoughtful feedback on my paper. I have made a number of changes to reflect your recommendations. Per your suggestion, I have removed the paragraph on DC statehood from my opening section, since I agree that it disrupts the flow of the introduction. I have also expanded the thesis statement in my introduction to make the scope of my argument clearer. Since the special issue in which this article will appear is focused on power and race, I hope that the relevance of my subject will be clear to my audience given the background I've presented. I have also added some clarification on why #dontmutedc protestors understood the muting of go-go as a racial and cultural issue. Given that this understanding was more or less universal among those who participated in the #dontmutedc protests, I have chosen not to theorize this understanding outside the history of the racialization of go-go that I have presented in the paper. Finally, I have acted on both your stylistic recommendations: I removed two of the "itselfs" between lines 233 and 234, and I switched each of the organization names to subheadings. Again, thank you for your feedback, which has helped me improve my paper.